# The Osteocyte: From “Prisoner” to “Orchestrator”

**DOI:** 10.3390/jfmk6010028

**Published:** 2021-03-17

**Authors:** Carla Palumbo, Marzia Ferretti

**Affiliations:** Section of Human Morphology, Department of Biomedical, Metabolic and Neural Sciences, University of Modena and Reggio Emilia, 41124 Modena, Italy; marzia.ferretti@unimore.it

**Keywords:** osteocytes, bone mechano-sensor, skeletal homeostasis, mineral homeostasis, bone remodeling

## Abstract

Osteocytes are the most abundant bone cells, entrapped inside the mineralized bone matrix. They derive from osteoblasts through a complex series of morpho-functional modifications; such modifications not only concern the cell shape (from prismatic to dendritic) and location (along the vascular bone surfaces or enclosed inside the lacuno-canalicular cavities, respectively) but also their role in bone processes (secretion/mineralization of preosseous matrix and/or regulation of bone remodeling). Osteocytes are connected with each other by means of different types of junctions, among which the gap junctions enable osteocytes inside the matrix to act in a neuronal-like manner, as a functional syncytium together with the cells placed on the vascular bone surfaces (osteoblasts or bone lining cells), the stromal cells and the endothelial cells, i.e., the bone basic cellular system (BBCS). Within the BBCS, osteocytes can communicate in two ways: by means of volume transmission and wiring transmission, depending on the type of signals (metabolic or mechanical, respectively) received and/or to be forwarded. The capability of osteocytes in maintaining skeletal and mineral homeostasis is due to the fact that it acts as a mechano-sensor, able to transduce mechanical strains into biological signals and to trigger/modulate the bone remodeling, also because of the relevant role of sclerostin secreted by osteocytes, thus regulating different bone cell signaling pathways. The authors want to emphasize that the present review is centered on the morphological aspects of the osteocytes that clearly explain their functional implications and their role as bone orchestrators.

## 1. Osteoblast-to-Osteocyte Transformation

It has been known for more than a century [1,2,3] that the osteocyte originates from the osteoblast. However, the process of osteoblast-to-osteocyte differentiation has been widely investigated only at a later time point with regard to both morphological and functional aspects. The structural differences between osteoblasts and osteocytes were shown by various authors in the 1960s to1980s [4,5,6,7,8], but only afterwards was established the temporal sequence of the events that allows the transformation of the prismatic osteoblast (generally arranged in laminae facing the vascular bone surfaces) into the dendritic mature osteocyte (embedded in the mineralized matrix) [9,10,11]. Concerning the dynamic modification of the cell body of preosteocyte (i.e., the differentiating osteocyte), the amount of the cytoplasmic organelles decreases, whereas the nucleus-to-cytoplasm ratio increases depending on the diminution of its secretive activity [9]. In parallel to both the cellular body reduction in size and the modification in ultrastructure, the formation of the cytoplasmic processes proceeds with an asynchronous and asymmetric pattern, considering that the cells in differentiation are progressively further away from the vascular surface due to the osteoid secreted by the osteoblastic lamina (Figure 1): firstly, the differentiating osteocytes maintain contacts with the mature osteocytes that recruited them from the osteoblastic lamina, forming short “mineral” processes; later, they establish contacts with the migrating osteoblastic lamina, elongating slender and long “vascular” processes, issued before the complete mineralization of the surrounding osteoid. The asynchronous and asymmetrical dendrogenesis is the expression of the fact that osteocytes (as all bone cells) live in an asymmetrical environment, between the mature mineralized matrix and the vascular surface (covered by osteoblastic laminae or bone lining cells); thus, it is logical to expect that not only the osteocytes, but also the preosteocytes and the osteoblasts, are morpho-functionally asymmetric cells.

At the end of the process, the osteocytes are confined to lacuno-canalicular cavities, “prisoners” inside the mineralized matrix. Despite this fact, they are connected, thanks to the dendrogenesis process, to each other and with the bone cells on the vascular surfaces through a network of dendrities, running within the canalicular network; this condition allows osteocytes to act as “orchestrators” of bone processes [12,13]. Prerequisite for that is the existence of junctional complexes occurring among osteocyte cytoplasmic processes [7,14,15], suggesting that the bone cells of the osteogenic lineage, arranged in network (Figure 2), might act as a functional syncytium, that includes also the cells covering the vascular bone surfaces, bone lining cells [15] or osteoblasts [16].

In conclusion, throughout the whole differentiation process, preosteocytes are always in close relationship with the neighboring cells (osteoblasts, osteocytes) by means of variously-shaped intercellular contacts (invaginated finger-like, side-to-side, and end to-end) and two types of specialized junctions: gap and adherens [14]. The pivotal role played by these contacts and junctions in osteocyte differentiation and activity will be discussed in the context of their distinct functional significance.

## 2. Osteogenesis Processes and Osteocyte Morphology

To fully understand the osteocyte morphology in the different contexts in which it originates, it is important to mention that morphological studies have shown that osteocyte differentiation is quite different in two types of osteogenesis. We clearly demonstrated, for the first time in 2002 in our labs, that not all osteoblasts are arranged in the well-known movable laminae: the two types of osteogenesis were named, during our studies on intramembranous ossification, “static bone formation” or static osteogenesis (SO) and “dynamic bone formation” or dynamic osteogenesis (DO) [17,18,19,20,21]. SO and DO occur in temporal sequence and depend on whether or not osteoblast movement occurs. We pointed out that the well-known dynamic osteogenesis (DO), performed by migrating osteoblast laminae, is preceded by static osteogenesis (SO) in which, at the onset of ossification, cords of stationary osteoblasts (instead of laminae of movable osteoblasts) transform into osteocytes in the same site where they differentiated from mesenchymal progenitor elements. In SO (Figure 3A,B), stationary osteoblasts (acting in sites where no bone pre-exists) are in very peculiar and unusual conditions with respect to the well-known dynamic osteoblasts, arranged in laminae and secreting bone close to the preexistent one: i.e., they are in a symmetrical environment since the osteoblast cords differentiate at about half distance between two vessels; in this extent, several factors are involved, such as endothelial-cell-derived cytokines (Endothelin-1) and growth factors (EDGF, PDGF). In fact, they give rise to big globous osteocytes, often located inside confluent lacunae, with short and symmetrical dendrites that can radiate simultaneously from all around their cell body because they are completely surrounded by the unmineralized matrix. In DO (Figure 3C,D), instead, osteoblasts transform into small ovoidal/ellipsoidal spidery osteocytes inside an asymmetrical environment, whose dendrites form in an asynchronous and asymmetrical manner in concomitance with, and depending on, the advancing mineralizing surface and the migrating osteogenic laminae.

Among all osteocytes (both SO- and DO-derived) and between osteocytes and osteoblasts, simple contacts and specialized gap junctions were observed, that allow all bone cells to remain functionally connected; therefore, a continuous osteocyte network extends throughout the bone, regardless of its static or dynamic origin. This cellular network has the characteristics of a functional syncytium, potentially capable of modulating, by wiring transmission (see below), the cells of the osteogenic lineage covering the bone surfaces. Morphology, as well as density and distribution of osteocytes, is not only related to the type of osteogenesis from which they derive but also to the collagen texture where they are located, on whose spatial organization the osteocytes play a pivotal role [22]. Unlike what was traditionally reported in textbooks in relation to bone histology, about 3 decades ago Marotti and collaborators reworked the classification of bone tissues, demonstrating, by means of both SEM (scanning electron microscopy) and TEM (transmission electron microscopy) analyses, that woven bone can be arranged in two different tissues [10,22,23,24]: “not lamellar woven bone” mostly formed by SO and “lamellar woven bone” mostly formed by DO. Comparing the two type of tissue (Figure 4): in not-lamellar-woven-bone (Figure 4A), osteocytes are symmetric, randomly distributed in clusters, with a large and irregularly globous shape of the cell body; in lamellar-woven-bone (Figure 4B) (made up of alternate dense and loose lamellae, both with an interwoven fibrous texture), osteocytes are asymmetric, distributed in planes exclusively located inside loose lamellae, with an almond-like shape of the cell body (i.e., a triaxial ellipsoid) [10,25,26,27,28]. It is to be underlined that, generally, morphology of osteocytes is indirectly inferred from the shape of the lacuno-canalicular network in which they are enclosed rather than directly from their protoplasm.

But we can ask: “was the egg or the hen born first?” that is to wonder if osteocyte affects the collagen texture or vice-versa the collagen texture modifies the cell morphology and arrangement? The fact that lamellar bone appears to be a variety of woven bone and that osteocytes are located in loose lamellae only has suggested that the differences in texture between lamellar-woven and not-lamellar-woven bone depend on the distribution of osteocytes throughout the bone matrix, that is to say, on the manner of recruitment of the osteocyte-differentiating osteoblasts from the osteogenic laminae [22,23,29]. In not-lamellar-woven-bone (generally laid down very rapidly) the osteoblasts that differentiate into osteocytes are recruited haphazardly and “enter” the bone in a random fashion. As a result, such type of not-lamellar-woven bone consists of an irregular distribution of osteocyte-rich areas where the collagen is loosely arranged (since it corresponds to that of the perilacunar matrices) (Figure 4A,C).

In lamellar-woven-bone, whose matrix is usually laid down at a lower rate than that of not-lamellar-woven-bone, the recruitment of the osteoblasts that differentiate into osteocytes occurs in an orderly manner. Since the cellular lamellae are only those formed by loose texture, this suggests that the osteoblasts committed to differentiating into osteocytes are recruited in successive groups and that those pertaining to each group are distributed in a single plane, namely that corresponding to a loose lamella. Thus, loose lamellae could simply form as a result of the alignment and fusion of the loosely-arranged fibers pertaining to the loose perilacunar matrices of the osteocytes they contain (Figure 4B,D).

Moreover, van Oers and coworkers [28] showed interactions between osteocyte shape and mechanical loading. In particular, their correlation seems to be related to the orientation of collagen fibers during osteoblast bone deposition: collagen gives bone its tensile strength which, in turn, conditions the elongation of osteocyte axes in relation to the direction of stress lines. These interactions are twofold: firstly, as regards the mechanical loading affecting the shape of osteocyte lacunae; secondly, as regards the shape of osteocytes influencing their mechano-sensibility and subsequent control of bone remodeling (see below). Shape variations of the lacuna hosting the osteocyte will also alter the direct cell strain, as well as the fluid flow and, in case, the microdamage inputs to the osteocyte. Furthermore, the shape of the osteocyte cell body affects its sensitivity to these inputs [30].

## 3. Osteocyte Network and Communications

Morphological investigations [31,32] showed that an osteogenic cellular system is present inside the bone, formed by various cell types from depth to the vascular surfaces: osteocytes, osteoblasts, or bone lining cells (according to whether the bone surfaces are in growing or resting phase, respectively), stromal cells, and endothelial cells. They constitute a functional syncytium whose cells play different roles and have different relationships with the surrounding environment. Osteocytes are enclosed inside bone microcavities filled with the bone fluid compartment (BFC); the stromal cells, extending from vascular endothelia to the cells carpeting the bone surfaces, are bathed by the perivascular interstitial fluid (PIF); osteoblasts and/or bone lining cells, located at the interface “mineralized matrix/vascular surfaces”, separate the two different fluid compartments and are in contacts both with the “deep” and “superficial” cell populations. Intercellular contacts (simple contacts and/or specialized junctions) were observed throughout all cells of the osteogenic system [11,14]. Thus, these cells form a continuous protoplasmic network that extends from the osteocytes to the vascular endothelia [33]. The presence of gap junctions (allowing electric coupling) suggests that the bone osteogenic cells could be considered a functional syncytium regulated not only by diffusion through the intercellular fluids (volume transmission) but also by signals issued through the cytoplasmic network and driven through cytoplasmic extentions (wiring transmission) (see below).

In the resting phase (namely, when no bone formation/erosion occur massively) osteocytes, bone lining cells, and stromal cells were named, as a whole, the bone basic cellular system (BBCS) (Figure 5) [31,32], because they represent the only permanent cellular background capable of perceiving mechanical strains and biochemical factors and then of triggering and driving both processes of bone formation and bone resorption.

It is appropriate to underline, in this context, that the majority of researchers consider osteoblasts and osteoclasts as the only relevant bony cells. Here, the authors want to emphasize that such a view is incorrect and misleading: indeed, osteoblasts and osteoclasts (the bone forming and bone reabsorbing cells, respectively) are transient cells, thus they cannot be the first to be involved, in the resting phase, in sensing skeletal requirements which modulate bone processes. Briefly, according to our view, osteoblasts and osteoclasts represent the “arms” of a worker; the actual “operation center” (i.e., the “brain”) is constituted by the cells of the osteogenic lineage in the resting phase (BBCS), particularly the osteocytes. (Figure 6) [34]. As shown by our previous studies, signaling throughout BBCS can occur by volume transmission (VT) and/or wiring transmission (WT) [31,32,35,36]. VT corresponds to the routes followed by soluble substances (hormones, cytokines, growth factors), whereas WT represents the diffusion of ionic currents along cytoplasmic processes in a neuron-like manner. It is likely that biochemical signals first affect stromal cells embedded in PIF and diffuse by VT to reach the other cells of BBCS, whereas mechanical agents are firstly sensed by osteocytes bathed by BFC inside the mineralized matrix and then issued throughout BBCS by WT (Figure 7).

It should be noted that besides WT, other similarities do exist between osteocytes and neurons. The short and thick mineral cytoplasmic processes of osteocytes resemble neuronal dendrites, whereas the long and slender vascular cytoplasmic processes are similar to neuronal axons. Transmission of signals through osteocytes seems to occur by gap junctions instead of synapses, though it has been recently shown that osteocytes produce typical neurotransmitters, such as nitric oxide [37,38,39,40] and prostaglandin E2 [38,41]. Additionally, about two decades ago we provided evidence that WT occurs along osteocytes in amphibian and murine cortical bone depending on loading [42,43], demonstrating that osteocytes and bone lining cells are at the origin of ionic currents, by operating as a cellular membrane partition which regulates the ionic flows between bone (BFC) and plasma (PIF). Such osteocyte ionic currents are constantly directed to the bone lining cells and stromal cells during the resting phase.

As regards the transmission of mechanical signals, both recent and less recent literature indicates the osteocyte as the main strain-sensitive bone cell [44,45,46,47,48,49,50,51,52,53,54,55,56,57,58]. We have shown [42,43] that shear stress-activated osteocytes are capable of steadily increasing and maintaining the basal current produced by the ionic fluxes (streaming potentials), which occur inside the lacuno-canalicular microcavities in response to pulsing mechanical loading (Figure 8). Briefly, the fact that all osteocytes take part in the formation of a potential osteocyte syncytium supports the view that mechanical signals throughout bone cells are mainly issued by wiring transmission since volume transmission does not need cell contacts in order to occur. A functional syncytium as such is able to initiate, perform and stop bone remodeling in order to meet current metabolic and mechanical bone requirements. In this context, the osteocyte role, played in bone adaptation to mechanical stimuli, is well explained by the “Mechanostat Setpoint Theory” (Figure 9), according to which the osteocytes are likely capable of modulating bone resorption and bone formation within a range of physiological mechanical strains whose upper and lower limits are named setpoints. When the lower (about 50 µ∑) and upper (about 2500 µ∑) setpoint values are exceeded bone resorption and bone formation, respectively, are triggered [44,59,60,61].

## 4. Osteocytes as Bone Mechanical Sensors and Transducers of Mechanical Strains into Biological Signals

Lozupone and coworkers [62] showed in vitro that the number of gap junctions between osteocytes increases in answer to the mechanical load applied to bone segments placed in cultures. This is in line with the suggestion that gap junctions play an important role both in cell-to-cell communications and in cell synchronization, enabling small molecules to be exchanged between the coupled cells [11,14,15,16,63]; it is also in line with the fact that osteocytes are considered as mechanoreceptors, able to give rise to the transduction mechanical strains into biological stimuli. LM data also demonstrated that the number of viable osteocytes is higher in loaded bones than in the control unloaded ones; TEM analyses showed that the intermittent compressive loading not only exerts a trophic function on osteocytes, probably improving the fluid flows inside the canalicular network [64], but also stimulates their metabolic activity, particularly their protein and/or glycoprotein synthesis, as shown by the increment of rough endoplasmic reticulum, thus suggesting that these molecules are secreted in answer to mechanical stimuli.

Moreover, Rubinacci and coworkers [42] showed that the damage-generated ionic currents they observed in the cortex of frog metatarsal bones had a cellular origin. Metatarsal bones of adult amphibian were purposely selected for the study because their shafts (unlike the mammalian one) contain only the osteocyte-bone lining cell system, thus giving the first convincing evidence that damage-generated ionic currents are generated by the osteocyte-bone lining cell system, i.e., the actual responsible “entity” of bone adaptation to both mechanical loading and mineral homeostasis. Later, the same authors [43] confirmed the observations also in murine models.

Recently [58], as far as the bone adaptation-related cell-signaling is concerned, both Erk1/2 and Akt were showed to be hyperphosphorylated in frog long bones of stressed samples (forced swimming) suggesting that among the putative osteocyte signal transduction mechanisms, Akt signaling is boosted by increased mechanical stresses. Moreover, the authors confirmed again that the increase of osteocyte gap junction number is dependent on mechanical loading (Figure 10).

## 5. Bone Remodeling and Sclerostin Involvement

The bone remodeling process is characterized by distinct phases [65,66] (Figure 11). During unloading, or when sensitivity to strain is altered by hormones (such as parathyroid hormone, estrogens, etc.), osteocytes stop producing the ionic currents maintaining the resting phase, thus allowing the bone lining cells, the stromal cells and, above all, the osteocytes (the most sensitive ones to loading changes) to produce RANKL, that activates the osteoclastogenesis (first phase resorption). During bone resorption, BBCS is destroyed, but it is likely that surviving overstrained osteocytes are involved in crucial signaling that stops osteoclast activity and triggers the reversal phase (second phase reversion). Various authors believe that the cells of the reversal phase could be involved in sending or receiving these signals [67,68,69]. Other signaling pathways may include matrix-derived factors such as bone morphogenic protein (BMP)-2, transforming growth factor (TGF)-β, and insulin-like growth factor (IGF) [66,70,71]. The cells of the reversal phase (probably of stromal-fibroblast origin) differentiate into osteoblasts, thus bone formation occurs (third phase deposition). During bone deposition, BBCS is progressively renewed and osteoblast activity stops when local physiological loading conditions are restored; the osteocytes in the newly-laid-down bone matrix produce again the steady ionic currents that return the bone to the resting phase, therefore halting osteoblast activity.

As far as the regulation of bone remodeling process is concerned, it has recently been shown that sclerostin has a relevant role, displaying both autocrine and paracrine effects [66]. Sclerostin (a 22-kDa glycoprotein), due to SOST promoter hypomethylation [72,73], is currently considered the major mediator of the molecular osteocyte mechanisms involved in the process of adaptive bone responses. In the mature skeleton, sclerostin is mainly synthesized by differentiated mature osteocytes, while preosteocytes, bone lining cells, and osteoblasts express very low levels of sclerostin. The central role of sclerostin is performed through the interplay between two opposing mechanisms: (1) unloading-induced-high sclerostin levels, that antagonize canonical Wnt in osteocytes and osteoblasts and promote non-canonical Wnt and/or other pathways in osteocytes and osteoclasts [74,75,76]; (2) mechanical loading-induced-low sclerostin levels, that activate Wnt-canonical signaling and bone formation. Thus, adaptive bone remodeling, occurring in different bone compartments, is driven by altered sclerostin levels, which regulate the expression of the other osteocyte-specific proteins, such as RANKL, its decoy receptor osteoprotegerin (OPG), and other proteins (Figure 12). Under the regulation of differential RANKL and OPG production, due to specific conditions, sclerostin creates a dynamic RANKL/OPG ratio [77,78,79], leading to either bone resorption or formation. Such opposite up- or down-regulation of the remodeling phases allows osteocytes (i.e., the permanent bone cells) to function as “the orchestrators” of osteoclasts and osteoblasts (i.e., the transient operating bone cells) ensuring the transition from bone resorption to bone formation. In this context, the putative inhibition of sclerostin represents a strategy to target bone remodeling unbalance in skeletal disorders, as osteoporosis, rheumatoid arthritis, bone related-genetic disorders, notwithstanding further studies are needed to better clarify how sclerostin-Ab modulates bone resorption since some authors reported the lack of modulation of RANKL, OPG, and other regulators of osteoclastogenesis during sclerostin-Ab treatment [80,81,82]. Recently, various papers were published on the topic where new therapeutic strategies were proposed, as denosumab, rosomozumab, etc. [83,84,85,86].

## 6. Interplay between Mineral and Skeletal Homeostasis and Osteocyte Role Mediated by Sclerostin

Bone remodeling is the main tool by which the skeleton answers to both the metabolic and mechanical demands, thus regulating the mineral and skeletal homeostasis. The mineral homeostasis keeps in relatively stable balance the concentration of mineral ions, as calcium and phosphate, in the organic liquids; the skeletal homeostasis allows the adjustment of shape, mass and bone structure following the actual mechanical needs of the skeletal segments. The two processes are not independent of each other; in fact, various investigations showed that they are functionally correlated in driving the bone responses, as a whole, to different experimental conditions [87,88,89,90,91].

Here, the authors want to stress the importance of the interplay between mineral and skeletal homeostasis (i.e., a dynamic balance of the two homeostases) in modulating and guiding bone’s response to dietary/metabolic alterations and/or unbalance of loading conditions. In particular, it is important to be emphasized that mineral homeostasis involves bone response to a variation of mineral serum levels (in consequence of various conditions, such as hormonal alterations or dietary regimen), while skeletal homeostasis implies bone answers to loading modifications.

In metabolic osteoporosis due to a calcium-deprived diet for one month in a rat model, Ferretti and coworkers [86] showed that the lack of calcium in the diet does not lead to a unique bone answer, since the interplay between mineral and skeletal homeostasis influence the bone loss in different sites of the two bony architectures (trabecular versus cortical bone) in both axial (lumbar vertebrae) and appendicular (femurs) skeleton. Despite the observed reduction of trabecular number (due to the maintenance of mineral homeostasis), an intense activity of bone deposition occurs on the surface of the few remaining overloaded trabeculae (in answering to mechanical stresses and, consequently, to maintain the skeletal homeostasis). The authors also reported: (i) the evidence that the more involved bony architecture is the trabecular one, that is the most susceptible to dynamic homeostasis variations; (ii) the different responses recorded in different sites of cortical bone are dependent on their main function in answering mineral and/or skeletal homeostasis.

Moreover, on the basis of evidence that PTH(1-34) improves recovery of bone fragility and accelerates bone healing [92,93,94,95,96,97,98], a recent study, performed by Ferretti and coworkers in animal models [91], showed that, after one month of calcium-free feeding, the normal diet restoration with/without concomitant PTH (1-34) administration differently determines the pattern of bone mass recovery, in terms of amounts and sites, not only depending on the mineral homeostasis but also under the influence of the skeletal homeostasis. This study highlighted (i) the importance of the interplay between mineral and skeletal homeostasis, (ii) the evidence that the more involved bony architecture, in modulating and guiding bone’s response to dietary/metabolic alterations, is again the trabecular bone, (iii) the trabecular bone as the preferential target of PTH (1-34). These observations agree with those of various authors [99,100,101] showing that appendicular skeleton (mostly concerning cortical bone) answers mainly to mechanical demands (i.e., is devoted to the skeletal homeostasis) while axial skeleton (mostly concerning trabecular bone) answers mainly to metabolic demand (i.e., is mainly devoted to the mineral homeostasis).

With regard to the osteocyte’s role in mediating the interaction between the two types of homeostasis, it represents (as previously reported) both the bone mechano-sensor and the major producer of some signaling proteins. As previously mentioned, one of the most investigated osteocyte proteins is sclerostin, that inhibits osteoblast activity, depending (among other factors/conditions) on the answers to mineral and skeletal homeostasis. In this context, it is not surprising that sclerostin expression is higher in animals fed a calcium-free diet with respect to all animals with restored normal diet and to the control groups [90], also depending on the answers to mineral and skeletal homeostasis [102].

## 7. Conclusions

In conclusion, despite the “segregation” within the mineralized bone matrix, the osteocyte is the dynamic bone cellular element that triggers/guides/modulates a series of sophisticated and interconnected processes, in hierarchic priority: their balance is allowed by osteocyte capability to sense the different bone demands (i.e., metabolic, hormonal, mechanical, etc.) and, depending on the interaction with the actual systemic conditions, to act on the “operator” bone cells that form and destroyed bone tissue under the direction of the best orchestrator.

## Figures and Tables

**Figure 1 jfmk-06-00028-f001:**
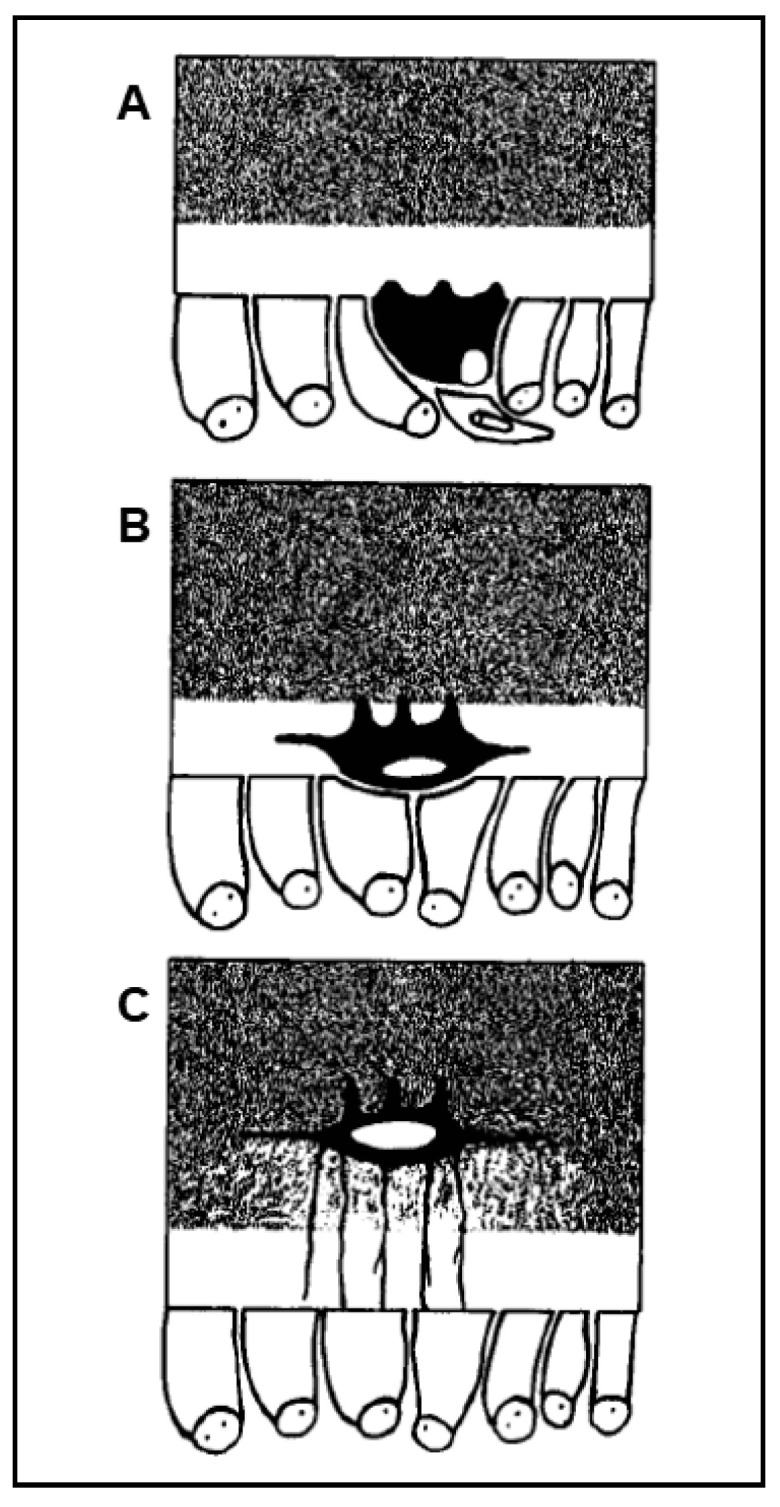
Schematic drawing showing the asynchronous and asymmetric pattern of cytoplasmic processes formation during osteocyte differentiation (preosteocytes = black; osteoblasts = white). (**A**) Preosteocyte enlarges its secretory territory, thus reducing its appositional growth rate, and starts to radiate processes towards the osteoid. (**B**) Preosteocyte, located inside the osteoid seam but still in contact with the osteoblastic lamina, continues to radiate short and thick cytoplasmic processes only from its mineral-facing side. (**C**) Preosteocyte, before being completely buried by minerals, radiates long and slender processes from its vascular-facing side to remain in touch with the osteoblastic lamina.

**Figure 2 jfmk-06-00028-f002:**
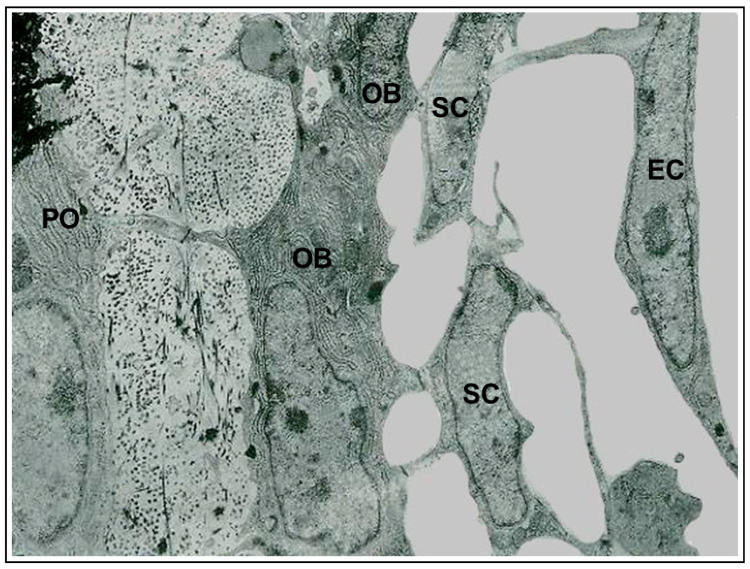
Transmission electron microscope (TEM) micrograph showing the continuous cytoplasmic network of the cells of the osteogenic lineage, extending from osteocytes to endothelial cells. (PO preosteocyte, OB osteoblast, SC stromal cell, EC endothelial cell). ×22,500.

**Figure 3 jfmk-06-00028-f003:**
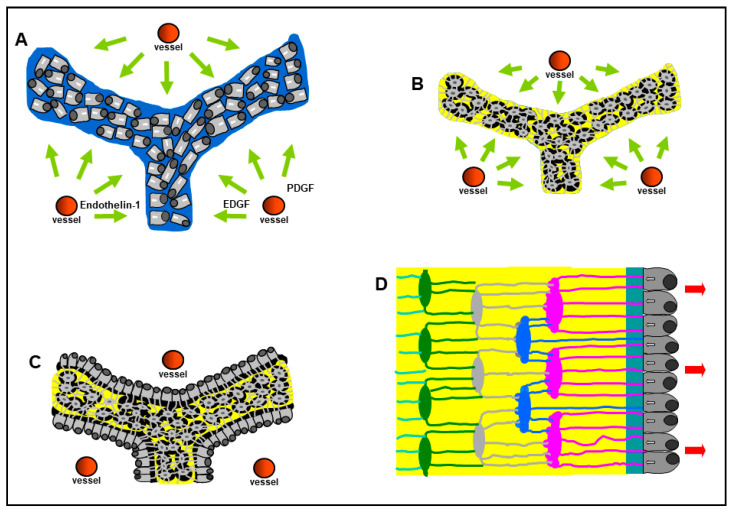
Schematic drawing showing the occurrence, in sequence, of static (SO) and dynamic (DO) osteogenesis. In SO, stationary osteoblast cords (**A**) differentiate at about half distance between two vessels, giving rise to big globous osteocytes (**B**), often located inside confluent lacunae, with short and symmetrical dendrites. In DO, osteoblastic laminae differentiate on the surface of SO-derived trabeculae (**C**) and transform into small spidery osteocytes (**D**) whose asymmetrical dendrites form in concomitance with the migration of the osteogenic lamina. See text for explanation. (EDGF: endothelial-derived growth factor; PDGF: platelet-derived growth factor).

**Figure 4 jfmk-06-00028-f004:**
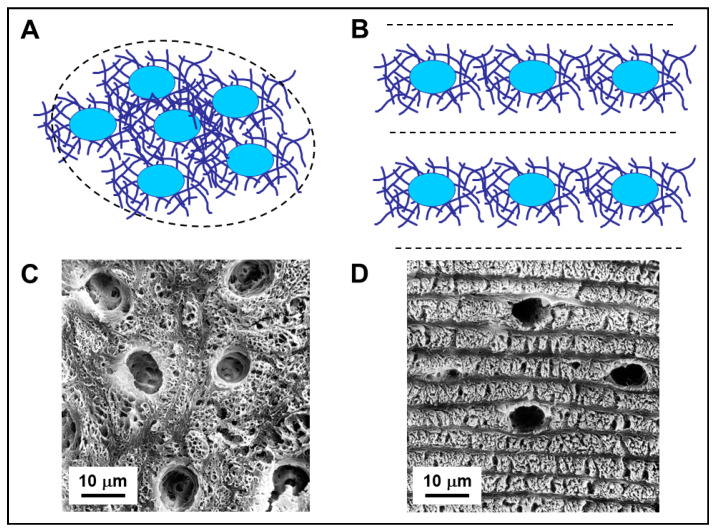
(**A**,**B**): Schematic drawing showing arrangement of osteocytes and the surrounding collagen fibers in not-lamellar-woven-bone (**A**) and in lamellar-woven-bone (**B**). In both types of bone tissue, osteocyte lacunae (blue ovals) are surrounded by perilacunar matrices of loosely arranged collagen fibers. In lamellar-woven-bone, the loose lamellae result as a consequence of alignment and fusion of the perilacunar loose matrices of the osteocytes arranged in planes; the dense bundles of collagen fibers do not contain osteocytes. (**C**,**D**): SEM micrographs of not-lamellar-woven-bone (**C**) and lamellar-woven-bone (**D**); the osteocyte lacunae are larger, more numerous, and irregularly distributed in not-lamellar-woven-bone with respect to lamellar-woven-bone, where they are only located inside loose lamellae. Note that dense lamellae, alternating with the loose ones, are thinner and do not contain osteocyte lacunae.

**Figure 5 jfmk-06-00028-f005:**
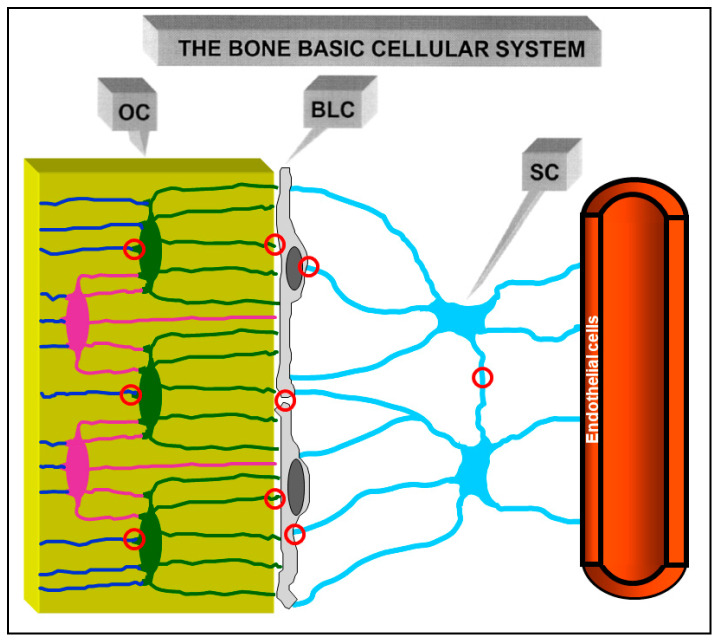
Schematic drawing of the cells of the osteogenic lineage in the resting phase, the Bone Basic Cellular System. From left to right: osteocytes (OC), bone lining cells (BLC), stromal cells (SC) and one vascular capillary. This cell network forms a functional syncytium since cells are joined by gap junctions (red circles).

**Figure 6 jfmk-06-00028-f006:**
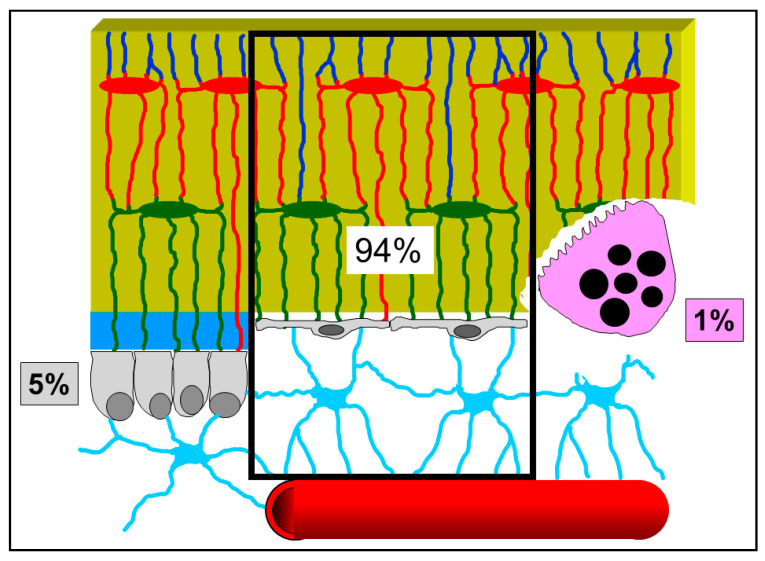
Schematic drawing showing the percentage of the various bone cells present in the bone resting phase: 94% BBCS, 5% osteoblasts, and 1% osteoclasts (data from Parfitt [34]). See text for explanation.

**Figure 7 jfmk-06-00028-f007:**
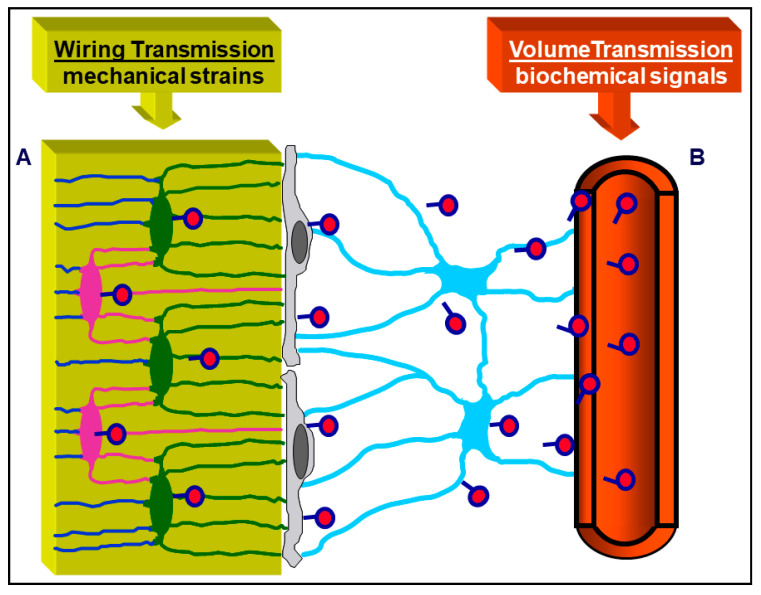
Schematic picture of the BBCS, in which signal transmissions can occur by volume transmission (VT) and/or wiring transmission (WT): (**A**) VT corresponds to the routes followed by soluble substances; (**B**) WT represents the diffusion of ionic currents along cytoplasmic processes. Biochemical signals (hormones diffusing from the blood, cytokines, etc.) first affect stromal cells and diffuse by VT to reach the other cells of BBCS, whereas mechanical agents are firstly sensed by osteocytes and then issued throughout BBCS by WT.

**Figure 8 jfmk-06-00028-f008:**
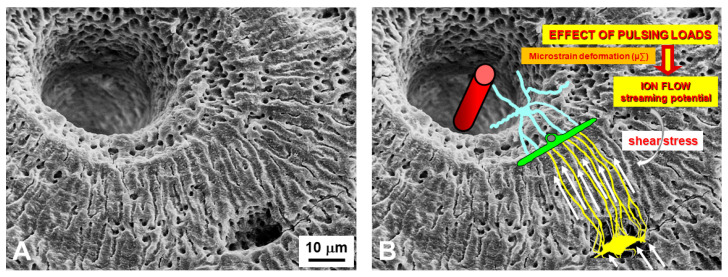
SEM micrograph of a portion of an osteon (**A**) showing the lacuno-canalicular cavities, on which the bone cellular system has been superimposed (in (**B**)), to explain the events occurring during mechanical stress: in response to mechanical loading, bone micro-deformations (induced by loading on bone) produce ionic fluxes (streaming potentials) in the bone fluid compartment that, in turn, activate the osteocytes via shear stress. Osteocytes, by means of wiring transmission, issue signals to the cells covering the bone surface, thus activating osteoblast recruitment and bone formation. Vessel (red cylinder inside the Haversian canal); stromal cell (light blue); bone lining cell (green); osteocyte (yellow).

**Figure 9 jfmk-06-00028-f009:**
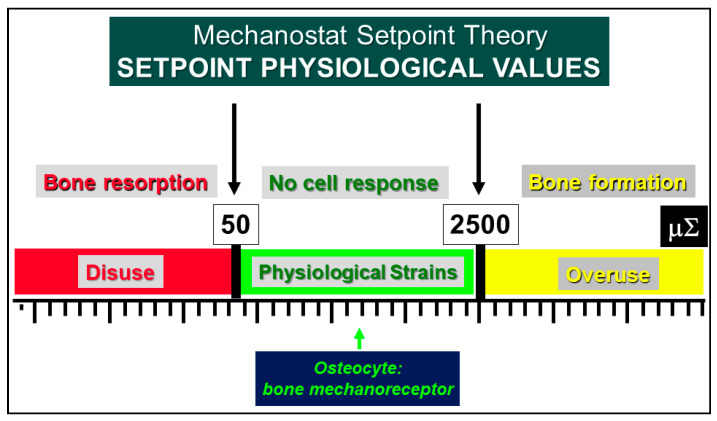
Schematic illustration of Frost’s Mechanostat Setpoint Theory. The setpoint values are expressed in microstrains (µ∑) and the bone mechanoreceptors (osteocytes) are activated only when the physiological setpoints are exceeded. In the “disuse” window, bone is lost owing to increased resorption. In the “overuse” window, bone is gained owing to increased bone formation.

**Figure 10 jfmk-06-00028-f010:**
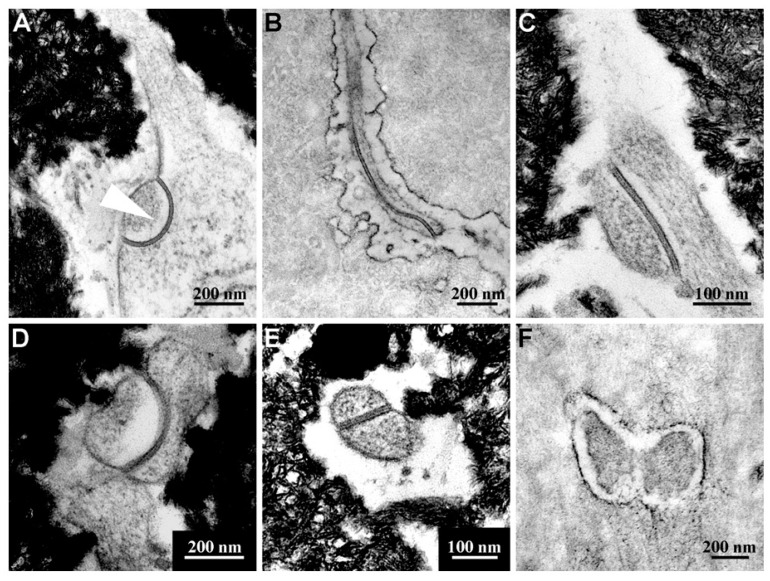
TEM micrograph panel representative of the abundance of gap junctions (**A**–**E**) with respect to simple contacts (**F**) in overloaded frogs. The head-arrow in (**A**) indicates the tip of a cytoplasmic process protruding in a cell body of an adjacent osteocyte showing an “invaginated finger-like” junction. Samples in (**B**) and (**F**) are decalcified.

**Figure 11 jfmk-06-00028-f011:**
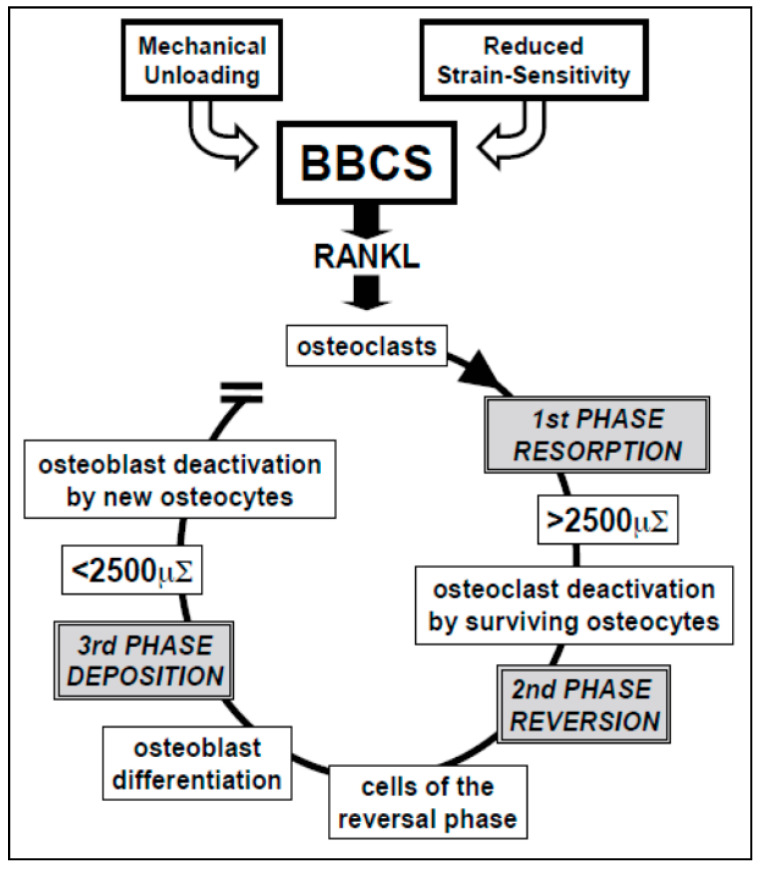
Schematic drawing of postulated BBCS role during bone remodeling cycle. See text for explanation.

**Figure 12 jfmk-06-00028-f012:**
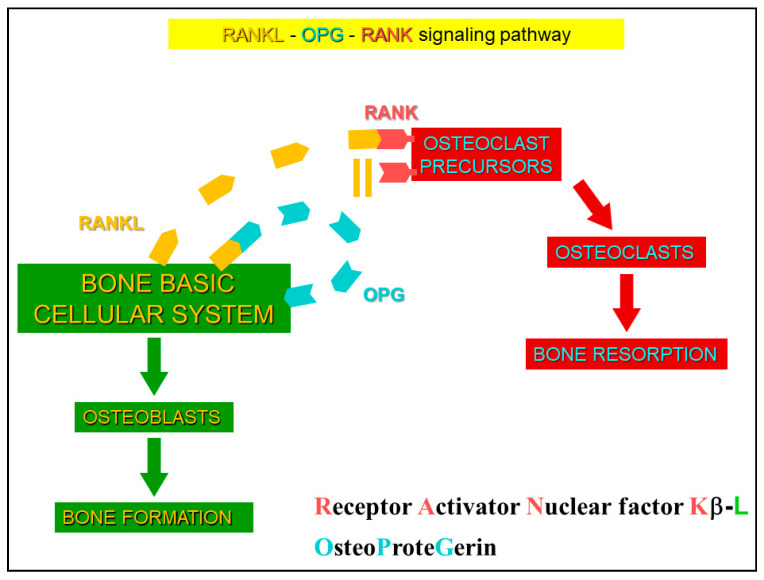
Scheme of RANK/RANKL/OPG system. Under specific conditions, differential RANKL and OPG production are induced, and the dynamic RANKL/OPG ratio leads to either bone formation or bone resorption.

## Data Availability

Data contained within the review are available in the papers cited in the section References.

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
