# Peer review of "The Osteocyte: From “Prisoner” to “Orchestrator”"

_jfmk, 2021, doi:10.3390/jfmk6010028_

Round 1

Reviewer 1 Report

This is a very comprehensive review of the interplay between the morphological and functional aspects of osteocyte biology by two authors who have made significant contributions to this field. The strength of the review lie in the depth analysis of osteocyte differentiation, communication networks, and role in remodeling, which is further higlighted by a number of very well designed illustrations as well as SEM/TEM images. On the other hand, the manuscript can be rather hard to follow at some parts because of long/complicated phrases.

A number of issues that should be adressed to as well as some suggestions can be found below:

Major points:

Page 1, line 42-43: The phrase ‚depending on the diminution of the secretive activity of the osteoblast‘ is difficult to follow-please clarify.

Page 2, lines 51-53: What do the authors mean by’ all bone cells live in an asymmetrical environment with regard to the vascular conditions‘? Please clarify.

Page 3, lines 97-100: It is not clear why the conditions of static osteogenesis are described as ‚peculiar and unusual‘?

Page 3, lines 97-100: It is unclear whether the growth factors mentioned only play a role in static, or also in dynamic osteogenesis?

Page 5, lines 154-160: Please explain the series of events in not-lamellar-woven bone better; this is a very long sentence and hard to follow.

Page 5-6: the part of the influence of mechanical loading on the osteocyte shape should be further elaborated upon.

Page 6, lines 202-204: please tone-down these statements

Page 8, lines 254-259: Very large sentence and the mechanostat setpoint values appear here out of the blue; please provide some context.

Page 10, lines 287-294: What is the physiological and pathophysiological role of the damage-generated ionic currents?

Page 12, lines 346-349: Monoclonal antibodies against sclerostin (i.e. romosozumab) are now widely used into clinical practice because of their efficacy in terms of bone mass accrual and anti-fracture activity especially in the setting of postmenopausal osteoporosis. Please rephrase and update literature to include relevant clinical studies.

Minor/Stylistic points:

Abstract: Osteocytes/ Osteocyte: for the sake of consistency please use either singular or plural form (and adjust grammar respectively) throughout the Abstract.

Abstract, pg 1, line 12:  delete ‚one‘ after ‚dendritic‘

Abstract, pg 1, line 14:  replace ‚the role playing‘ with ‚their role‘

Abstract, pg 1, line 16: rephrase to read :‘the gap junctions enable osteocytes inside the matrix to act‘

Abstract, pg 1, line 23:  change ‚thanks to‘ to ‚because of‘

Abstract, pg 1, line 24:  delete ‚the‘ before ‚different bone cell‘

Page 1, line 34: rephrase to read: ‚investigated only at a later timepoint with regard to both morphological and functional aspects‘.

Page 1, line 36: insert ‚was‘ before ‚the temporal sequence‘

Page 1, line 39: delete ‚was’ before ‚established‘

Page 2, line 51: insert ‚an‘ before ‚expression‘

Page 2, line 52: change to read ‚with regard to the vascular conditions‘

Page 2, lines 64-68: very long sentence-suggest to split in two sentences.

Page 2, line 64: change ‚notwithstanding‘ to ‚despite the fact that‘

Page 3, line 80: should be spelled ‚adherence‘

Page 3, line 82: change ‚peculiar‘ to ‚distinct‘

Page 3, lines 84-90: very long sentence-suggest to split in two sentences.

Page 3, line 117: insert ‚to remain‘ before ‚functionally connected‘

Page 3, line 118: should be ‚characteristics‘

Page 3, line 123: insert ‚was‘ before ‚traditionally‘

Page 5, line 163: ‚this suggests‘

Page 5, line 169: ‚showed‘ instead of ‚show‘

Page 6, line 180: insert ‚and‘ before ‚endothelial cells‘

Page 6, lines 200-201: ‚capable of perceiving‘ and ‚triggering‘ and ‚driving‘

Page 8, line 234: ‚It should be noted‘

Page 8, lines 246-247: ‚both recent and less recent literature indicates that the osteocyte is the main…‘

Page 8, line 254: ‚a functional syncytium as such is able to initiate, perform and stop bone remodeling‘

Page 12, line 363: insert ‚of‘ after ‚independent‘

Page 12, line 370: insert ‚a‘ before ‚variation‘

Page 12, line 370: ‚conditions‘ instead of ‚condition‘

Page 13, line 392: Replace ‚in this study it emerged‘ with ‚This study highlighted‘

Page 13, line 400: ‚With regard to the osteocyte’s role‘

Page 13, line 402: ‚investigated proteins‘

Page 13, line 410: ‚triggers/guides/modulates‘

Author Response

Review 1

This is a very comprehensive review of the interplay between the morphological and functional aspects of osteocyte biology by two authors who have made significant contributions to this field. The strength of the review lie in the depth analysis of osteocyte differentiation, communication networks, and role in remodeling, which is further higlighted by a number of very well designed illustrations as well as SEM/TEM images.

On the other hand, the manuscript can be rather hard to follow at some parts because of long/complicated phrases.

Authors: Thank you for your appreciation.

A number of issues that should be adressed to as well as some suggestions can be found below:

Major points:

Page 1, line 42-43: The phrase ‚depending on the diminution of the secretive activity of the osteoblast‘ is difficult to follow-please clarify.

Authors: according to the reviewer’s request, the sentence was rephrased in the revised version of the manuscript.

Page 2, lines 51-53: What do the authors mean by’ all bone cells live in an asymmetrical environment with regard to the vascular conditions‘? Please clarify.

Authors: according to the reviewer’s request, the sentence was rephrased in the revised version of the manuscript.

Page 3, lines 97-100: It is not clear why the conditions of static osteogenesis are described as ‚peculiar and unusual‘?

Authors: according to the reviewer’s request, the sentence was rephrased in the revised version of the manuscript.

Page 3, lines 97-100: It is unclear whether the growth factors mentioned only play a role in static, or also in dynamic osteogenesis?

Authors: In a very recently review on SO versus DO (see new reference n. 21, added in the revised version of the manuscript), the authors referred that SO is triggered only by inductive stimuli, since during SO the bone matrix is laid down without preexistent osteocytes (which generally drive bone deposition by dynamic osteoblasts). Thus, during SO osteoblast activity occurs notwithstanding the lack of osteocyte guidance. Only after SO, DO occurs by means of movable osteoblastic laminae, that conversely are guided by osteocytes (located inside the bony trabeculae preliminarly secreted by SO); also, dynamic osteoblasts are also capable of “sensing” the vessels (towards which they move) through the stromal cells located between the migrating osteoblastic laminae and the vascular endothelia.

It does not seem appropriate to the authors to burden the manuscript with details that have just been published concerning the two different types of osteogenesis and their respective conditioning factors (inductive factors for SO, mechanical factors for DO).

Ref. n. 21

Ferretti, M.; Palumbo, C. Static Osteogenesis versus Dynamic Osteogenesis: A Comparison between Two Different Types of Bone Formation. Appl. Sci. 2021, 11, 2025. https://doi.org/10.3390/app11052025

Page 5, lines 154-160: Please explain the series of events in not-lamellar-woven bone better; this is a very long sentence and hard to follow.

Authors: according to the reviewer’s request, the sentence was rephrased and shortened in the revised version of the manuscript.

Page 5-6: the part of the influence of mechanical loading on the osteocyte shape should be further elaborated upon.

Authors: according to the reviewer’s request, the sentence was rephrased and integrated with further considerations in the revised version of the manuscript.

Page 6, lines 202-204: please tone-down these statements

Authors: done. The sentence was replaced with a simple one.

Page 8, lines 254-259: Very large sentence and the mechanostat setpoint values appear here out of the blue; please provide some context.

Authors: The Review is right! According to the reviewer’s request, the sentence was rephrased and integrated with further considerations in the revised version of the manuscript.

Page 10, lines 287-294: What is the physiological and pathophysiological role of the damage-generated ionic currents?

Authors: Damage-generated ionic currents are representative of ion fluxes at bone/plasma interface. Since osteocytes and bone lining cells are at the origin of the currents, they operate as a cellular membrane partition, that regulates ionic flow between bone and plasma. Since strain-related adaptive remodeling could also depend on ionic characteristics and flow of the bone fluid through the osteocyte lacuna-canalicular network, the results (reported in ref. n. 42 of the present review) support the view that osteocyte and bone lining cells may constitute a functional syncytium involved in mineral homeostasis as well as in bone adaptation to mechanical loading.

It does not seem appropriate to the authors to burden the manuscript with peculiar details that have already been published, concerning the generation of ion fluxes and the role of osteocyte/bone lining cell system in maintaing such fluxes.

Page 12, lines 346-349: Monoclonal antibodies against sclerostin (i.e. romosozumab) are now widely used into clinical practice because of their efficacy in terms of bone mass accrual and anti-fracture activity especially in the setting of postmenopausal osteoporosis. Please rephrase and update literature to include relevant clinical studies.

Authors: thanks for suggesting an aspect to improve; according to the reviewer’s request, in the revised version of the manuscript a sentence was rephrased, another one was added, also including citations on clinical studies.

Minor/Stylistic points:

Abstract: Osteocytes/ Osteocyte: for the sake of consistency please use either singular or plural form (and adjust grammar respectively) throughout the Abstract.

Abstract, pg 1, line 12:  delete ‚one‘ after ‚dendritic‘

Abstract, pg 1, line 14:  replace ‚the role playing‘ with ‚their role‘

Abstract, pg 1, line 16: rephrase to read :‘the gap junctions enable osteocytes inside the matrix to act‘

Abstract, pg 1, line 23:  change ‚thanks to‘ to ‚because of‘

Abstract, pg 1, line 24:  delete ‚the‘ before ‚different bone cell‘

Authors: done. The Abstract was corrected according to the Review’s suggestions.

Page 1, line 34: rephrase to read: ‚investigated only at a later timepoint with regard to both morphological and functional aspects‘.

Page 1, line 36: insert ‚was‘ before ‚the temporal sequence‘

Page 1, line 39: delete ‚was’ before ‚established‘

Authors: done. In the first page the text was modified according to the Reviewer’s requests.

Page 2, line 51: insert ‚an‘ before ‚expression‘

Authors: done.

Page 2, line 52: change to read ‚with regard to the vascular conditions‘

Authors: according to a Reviewer’s previous suggestion “vascular conditions” was deleted.

Page 2, lines 64-68: very long sentence-suggest to split in two sentences.

Page 2, line 64: change ‚notwithstanding‘ to ‚despite the fact that‘

Authors: done. The sentence was divided in two sentences and “notwithstanding was replaced by “despite …” .

Page 3, line 80: should be spelled ‚adherence‘

Authors: the gergal term for that type of junction is “adherens junction”

Page 3, line 82: change ‚peculiar‘ to ‚distinct‘

Authors: done.

Page 3, lines 84-90: very long sentence-suggest to split in two sentences.

Authors: done. The sentence was divided in two sentences

Page 3, line 117: insert ‚to remain‘ before ‚functionally connected‘

Authors: done.

Page 3, line 118: should be ‚characteristics‘

Page 3, line 123: insert ‚was‘ before ‚traditionally‘

Authors: done.

Page 5, line 163: ‚this suggests‘

Page 5, line 169: ‚showed‘ instead of ‚show‘

Authors: done.

Page 6, line 180: insert ‚and‘ before ‚endothelial cells‘

Page 6, lines 200-201: ‚capable of perceiving‘ and ‚triggering‘ and ‚driving‘

Authors: done.

Page 8, line 234: ‚It should be noted‘

Page 8, lines 246-247: ‚both recent and less recent literature indicates that the osteocyte is the main…‘

Authors: done.

Page 8, line 254: ‚a functional syncytium as such is able to initiate, perform and stop bone remodeling‘

Authors: the requested modification was included in a rephrased sentence (by a previous comment).

Page 12, line 363: insert ‚of‘ after ‚independent‘

Page 12, line 370: insert ‚a‘ before ‚variation‘

Page 12, line 370: ‚conditions‘ instead of ‚condition‘

Authors: done.

Page 13, line 392: Replace ‚in this study it emerged‘ with ‚This study highlighted‘

Page 13, line 400: ‚With regard to the osteocyte’s role‘

Page 13, line 402: ‚investigated proteins‘

Page 13, line 410: ‚triggers/guides/modulates‘

Authors: done.

Reviewer 2 Report

This is a well-written review  article. This manuscript can be accepted for publication.

Author Response

The authors thank the Reviewer for the appreciation.

Reviewer 3 Report

This review is a nice discussion of the central roles that osteocytes play in regulation of bone cell signaling pathways. Although osteocytes receive scant attention relative to the more heavily studied osteoblasts and osteoclasts, their essential permanence in bone and ability to physically interact with other bone cell types suggests their functions as "orchestrators" of bone remodeling as described by the authors. The review is thorough, with emphasis mostly on how the unusual morphology of osteocytes facilitates their interactions with other bone cells, though it also includes a discussion of the some of the soluble factors such as sclerostin that these cells use to regulate behavior of other cells. In addition, the figures are numerous and are generally quite understandable and useful to the reader. The text is well-written with only some minor proofreading required. I have only a few minor suggestions/comments that may help improve the manuscript:

  • I found Figure 3 to be somewhat confusing at first. Greater labeling of structures or more description of figure elements in the legend would be useful. For example, the blood vessels could be labeled, and the factors emanating from these vessels (indicated by arrows) should be labeled in a larger or darker text since these labels are small and easily overlooked.
  • In lines 216/217 and the legend for Figure 7, it is stated that “mechanical agents are firstly sensed by osteocytes bathed by BFC inside the mineralized matrix then issued throughout BBCS by WT” (i.e. wiring transmission). Since the authors have defined WT as the diffusion of ionic currents along cytoplasmic processes (as opposed to volume transmission, which includes soluble substances such as hormones, cytokines, etc.), I’m wondering whether the use of WT to propagate mechanical signals might not be overstated here, since (as the authors point out) mechanical strain also is well known to induce production of soluble signaling factors such as sclerostin, NO, and others. It seems to me that both WT and VT play key roles in propagating mechanical signals by osteocytes.
  • Somewhat along those lines, I’m wondering whether the strain itself can be propagated from cell to cell, since the osteocytes are in physical contact with numerous cells and form a functional syncytium. Could it be that strain can be sensed between cells along the actin cytoskeletal network? Is there any evidence for this or speculation about this as a potential signaling mechanism?

Author Response

Review 3

This review is a nice discussion of the central roles that osteocytes play in regulation of bone cell signaling pathways. Although osteocytes receive scant attention relative to the more heavily studied osteoblasts and osteoclasts, their essential permanence in bone and ability to physically interact with other bone cell types suggests their functions as "orchestrators" of bone remodeling as described by the authors. The review is thorough, with emphasis mostly on how the unusual morphology of osteocytes facilitates their interactions with other bone cells, though it also includes a discussion of the some of the soluble factors such as sclerostin that these cells use to regulate behavior of other cells. In addition, the figures are numerous and are generally quite understandable and useful to the reader. The text is well-written with only some minor proofreading required.

Authors: Thank you for your appreciation.

I have only a few minor suggestions/comments that may help improve the manuscript:

I found Figure 3 to be somewhat confusing at first. Greater labeling of structures or more description of figure elements in the legend would be useful. For example, the blood vessels could be labeled, and the factors emanating from these vessels (indicated by arrows) should be labeled in a larger or darker text since these labels are small and easily overlooked.

Authors: in figure 3, following the suggestion, we have indicated the vessels with the label "vessel" and written in black color the growth factor

In lines 216/217 and the legend for Figure 7, it is stated that “mechanical agents are firstly sensed by osteocytes bathed by BFC inside the mineralized matrix then issued throughout BBCS by WT” (i.e. wiring transmission). Since the authors have defined WT as the diffusion of ionic currents along cytoplasmic processes (as opposed to volume transmission, which includes soluble substances such as hormones, cytokines, etc.), I’m wondering whether the use of WT to propagate mechanical signals might not be overstated here, since (as the authors point out) mechanical strain also is well known to induce production of soluble signaling factors such as sclerostin, NO, and others. It seems to me that both WT and VT play key roles in propagating mechanical signals by osteocytes.

Somewhat along those lines, I’m wondering whether the strain itself can be propagated from cell to cell, since the osteocytes are in physical contact with numerous cells and form a functional syncytium.

We thank the reviewer very much for the keenness of the observation. It is true that the mechanical stimulus induces the production of soluble signaling factors by osteocytes (sclerostin, NO, etc.), but it is the authors' belief that this step is probably subsequent to the mechanical activation of the deeper osteocytes in the bone (which perceive the deformations induced by mechanical loading in the mineralized matrix in which they are embedded) that immediately, by WT (electrical coupling between cells, therefore in very short time), can activate other osteocytes as well as cells on bone surfaces, thus expanding the number of cells able to respond to the load (also by VT). Let's say that WT has an initial role, after load sensing, to expand the bone response also for VT.

Conversely, it is obvious that diffusible metabolic signals from the vessels to the bone must necessarily follow VT.

We do not think to burden the present review with these aspects, since in-depth studies on WT versus VT are in progress in our laboratories.

We really thank the reviewer for pointing this out.

Could it be that strain can be sensed between cells along the actin cytoskeletal network? Is there any evidence for this or speculation about this as a potential signaling mechanism?

As reported in a paper cited in the review (Bertacchini et al., 2017), we showed ‘tethering’ structures within intracanalicular-pericytoplasmic spaces, in contact with the plasma membrane of osteocyte processes; the internal side of plasma membrane is in contact, in turn, with osteocyte cytoskeleton. The 'tethering’ structures could be involved in allowing osteocyte sensitiveness to micro-strains in triggering bone response to load. Moreover, these ‘tethering’ structures exert a resistance to loading-induced fluid flows, and the resulting drag force may be sensed at the cytoplasmic process membrane.

The authors thank again the Reviewer for his comments and, again, we propose not to modify the text, because we are deepening the topic on the transduction of mechanical stimuli into biological signals: 'tethering’ structures –osteocyte membrane plasma-cytoskeleton (actin filament, ect.) - activation of signal pathways - etc.

Round 2

Reviewer 1 Report

Many thanks for providing the updated version of the manuscript and the authors' detaile responses. In my opinion, the authors have adequately adressed all comments raised by the authors and performed suggested changes. In my opinion, the manuscript would now be eligible for publication. I only have a minor comment with regard to the sentence before Refs. [83-86].

This should be changed from "Recently, various papers were published on the topic where new therapeutic strategies were proposed, as denosumab, rosomozumab, etc. [83-86]" to "Romosozumab is a monoclonal antibody against sclerostin with a dual mode of action. It enhances bone formation and simultaneously suppresses bone resorption. A number of clinical studies have proven the efficacy of romosozumab in terms of bone mass accrual and anti-fracture activity in the setting of postmenopausal and male osteoporosis [83-86]."